# Polyester degradation by soil bacteria: identification of conserved BHETase enzymes in *Streptomyces*
Jo-Anne Verschoor ⓘ , Martijn R. J. Croese, Sven E. Lakemeier ⓘ , Annemiek Mugge, Charlotte M. C. Burgers, Paolo Innocenti, Joost Willemse ⓘ , Marjolein E. Crooijmans ⓘ , Gilles P. van Wezel ⓘ , Arthur F. J. Ram ⓘ & Johannes H. de Winde ⓘ ✉

The rising use of plastic results in an appalling amount of waste which is scattered into the environment. One of these plastics is PET which is mainly used for bottles. We have identified and characterized an esterase from *Streptomyces*, annotated as LipA, which can efficiently degrade the PET-derived oligomer BHET. The *Streptomyces coelicolor Sc*LipA enzyme exhibits varying sequence similarity to several BHETase/PETase enzymes, including *Is*PETase, *Tf*Cut2, LCC, PET40 and PET46. Of 96 *Streptomyces* strains, 18% were able to degrade BHET via one of three variants of LipA, named *Sc*LipA, *S2*LipA and *S92*LipA. *SclipA* was deleted from *S. coelicolor* resulting in reduced BHET degradation. Overexpression of all LipA variants significantly enhanced BHET degradation. All variants were expressed in *E. coli* for purification and biochemical analysis. The optimum conditions were determined as pH 7 and 25 °C for all variants. The activity on BHET and amorphous PET film was investigated. *S2*LipA efficiently degraded BHET and caused roughening and indents on the surface of PET films, comparable to the activity of previously described *Tf*Cut2 under the same conditions. The abundance of the *S2*LipA variant in *Streptomyces* suggests an environmental advantage towards the degradation of more polar substrates including these polluting plastics.

Petroleum-derived plastics are among the most useful and widespread synthetic polymers in the modern world. They are relatively easy and cheap to produce, extremely versatile and durable materials, which makes them remarkably attractive for a wide range of applications[1,2]. This has caused the demand and production of plastics to steadily rise to an estimated 400 million metric tons (Mt) produced per year as of 2022[3]. An estimated 79% of plastic waste has been mismanaged or landfilled[4]. In combination with this increasing output, the durability, and recalcitrance of these polymers make accumulation and pollution by plastics a major global environmental concern[5]. Polyethylene terephthalate (PET) is an extensively used plastic mainly used to produce bottles, food packaging, clothing, and films. The huge demand for PET makes it one of the most common plastics with 24 million tons being produced worldwide in 2022[6]. Today, PET waste management in most countries mainly consists of incineration and landfilling. These practices are causing detrimental effects to the environment, such as leaching and the release of toxic compounds[7]. An increasing number of countries have recycling systems in place to prevent environmental accumulation and make renewed use of PET waste. Despite such efforts, the recovery ratio is still approximately 30% in Europe and the US. Only a minor fraction, is used to manufacture new products, with substantial amounts of PET still entering the environment[8,9]. While plastic waste is accumulating in the environment, the adaptation and response of soil and aquatic microorganisms to plastic exposure is imminent. Microbial response and adaptation mechanisms may help battle plastic pollution and initiate remediation and recycling options. An important phylum of bacteria remaining to be investigated is the *Actinobacteria*. This phylum consists of various complex genera such as *Thermobifida, Rhodococcus,* and *Streptomyces*. Where the genus *Thermobifida* contains approximately twenty enzymes with depolymerizing activity for PET- or its constituent oligomer bis(2-hydroxyethyl) terephthalate (BHET), for various other genera including *Streptomyces* only limited information is available[10–15]. *Streptomycetes* are filamentous soil bacteria with a characteristic lifecycle. They are renowned for their capability to produce antibiotics but are also known to excrete enzymes to degrade complex natural polymers, including plant biomass[16–18]. Exposure to plastics and their ability to degrade complex biomass with various hydrolytic enzymes makes them a promising genus to explore for novel PET hydrolyzing enzymes (PHEs)[10–15].

Several promiscuous actinobacterial enzymes exhibiting BHET or PET-degrading activity have been identified such as the leaf compost

Molecular Biotechnology, Institute of Biology, Leiden University, Sylviusweg 72, 2333BE Leiden, The Netherlands. ✉e-mail: j.h.de.winde@biology.leidenuniv.nl

cutinase (LCC), *Tf*Cut2 from *Thermobifida fusca* and PET40, present in both *Amycolatopsis* and some *Streptomyces* species, earning them their place in the PAZY database[19-26]. For the entire genus of *Streptomyces*, only three PHEs have been described, namely SM14est, Sub1, and PET40[11,27,28]. Hence, limited data is available concerning the ability of *Streptomyces* to degrade BHET and PET and thus on the emergence, distribution, conservation, and activity of their corresponding BHET-degrading enzymes.

We have investigated the occurrence and conservation of BHET-degrading enzymes in *Streptomyces coelicolor* and a number of *Streptomyces* species, as an indication of emergence of PET-degrading activity in this genus. Via homology search using well-known PHEs *Is*PETase from *Ideonella sakaiensis*[29], LCC, *Tf*Cut2, and the recently identified PET40 we identified an enzyme annotated in *Streptomyces coelicolor* as Lipase A (*Sc*LipA), as a BHET-degrading enzyme. *Sc*LipA was previously annotated in a genomics study, however, it was never further characterized[30]. The presence of this BHET-degrading enzyme among other *Streptomycetes* was investigated by screening a collection of 96 previously isolated *Actinobacteria* (predominantly *Streptomyces*) under various conditions[31]. Overall, 18% of the evaluated strains were able to individually degrade BHET, confirming a wider distribution of this ability in nature. LipA is highly conserved in these strains and can be classified into three different variants. The functionality of all three LipA variants for degradation of BHET as well as of amorphous PET was investigated in vivo and in vitro.

In conclusion, we present a comprehensive identification and characterization of a novel, conserved BHET-degrading enzyme family of *Streptomyces*. Structural characteristics and in vivo and in vitro activity provide further insight into the evolutionary and ecological adaptation of soil bacteria towards plastics in their environment.

## Results
### Scavenging the *Streptomyces* genome for candidate PETase homologs
Homologs of the previously characterized PHEs, LCC, *Tf*Cut2, PET40, and *Is*PETase encoded by the genomes of *Streptomyces coelicolor*, *Streptomyces scabies,* and *Streptomyces avermitilis* were identified using the protein

BLAST. The most promising enzyme was an *S. coelicolor* enzyme, previously annotated as a putatively secreted lipase A (LipA, accession number Q9L2J6[30]), with a query coverage of 91% and highest identity of 78% with PET40 (Fig. 1a). Where the query coverage was similar for all investigated enzymes, the percent identity varied from 78% with PET40 to 47% with the *Is*PET*ase* (Fig. 1a). An AlphaFold 2 3D model[32] of *Sc*LipA and the 3D structure of PET40 (Fig. 1b) appeared highly similar[33]. The catalytic triad was fully conserved, as well as two of the three residues of the predicted PET substrate binding site. The first binding residue is identical to the one of PET40 and shows minor variation between previously described PETases[34]. The high similarity and the presence of previously described important residues for PET degradation hint towards the putative PET-degrading activity of the *Streptomyces coelicolor* lipase A (*Sc*LipA). On the N-terminal end of the protein, a twin-arginine translocation (TAT) signal is present indicating that the natural substrate of this enzyme is located outside of the cell[35]. Indeed, for plastics-degrading enzymes, extracellular activity appears to be essential.

### BHET-degrading activity of *S. coelicolor*
As a first investigation into the putative activity of *S. coelicolor* M145 on PET and derived oligomers, the strain was grown on *Streptomyces* minimal medium (StrepMM) agar plates containing BHET. Due to its limited solubility, the addition of bis(2-hydroxyethyl) terephthalate (BHET) causes the solid growth medium to become turbid. When BHET is converted to mono-(2-hydroxyethyl) terephthalic acid (MHET) and/or terephthalic acid (TPA), a halo of clearance appears. Secreted enzyme activity was expected to be tightly regulated in response to environmental growth conditions[36]. We anticipated that *N*-acetyl glucosamine (GlcNAc) may act as an inducer for the production of extracellular enzymes such as lipases, and esterases as well as putative BHET-degrading enzymes. GlcNAc is a monomer of chitin and peptidoglycan, which under poor nutritional conditions induces development and antibiotic production in *Streptomyces coelicolor*[37,32], whereas in nutrient-rich environments these responses are blocked[36]. Therefore, we tested GlcNAc as a possible inducer of secreted enzyme activity.

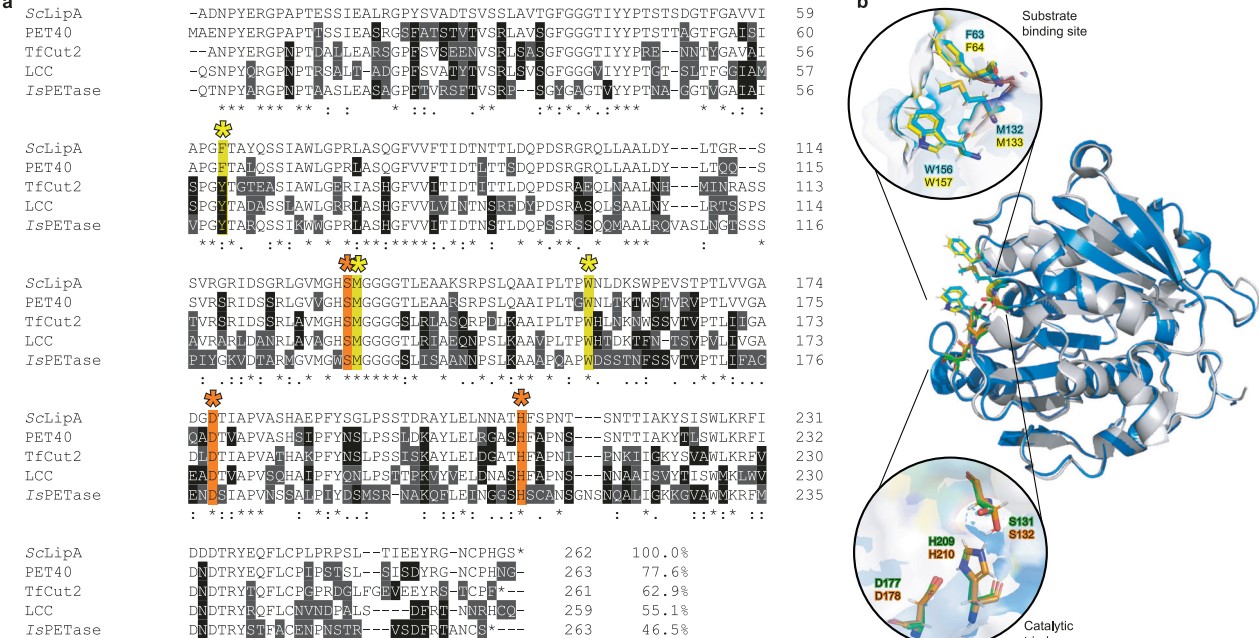

**Fig. 1 | Sequence comparison of *Sc*LipA with several PHEs. a** Sequence comparison of *Sc*LipA with PET40, *Tf*Cut2, LCC, and the *Is*PETase indicating the binding sites with yellow asterisks and highlights and the catalytic triad highlighted in orange with orange asterisks. The percent identity was displayed behind the alignments. **b** Predicted model of the structure of *Sc*LipA constructed with AlphaFold (blue) overlayed with the structure of PET40 as provided by PDB (8A2C). The catalytic triads are enlarged and shown in green (*Sc*LipA) and orange (PET40). The enlarged binding domains are displayed in cyan (*Sc*LipA) and yellow (PET40).

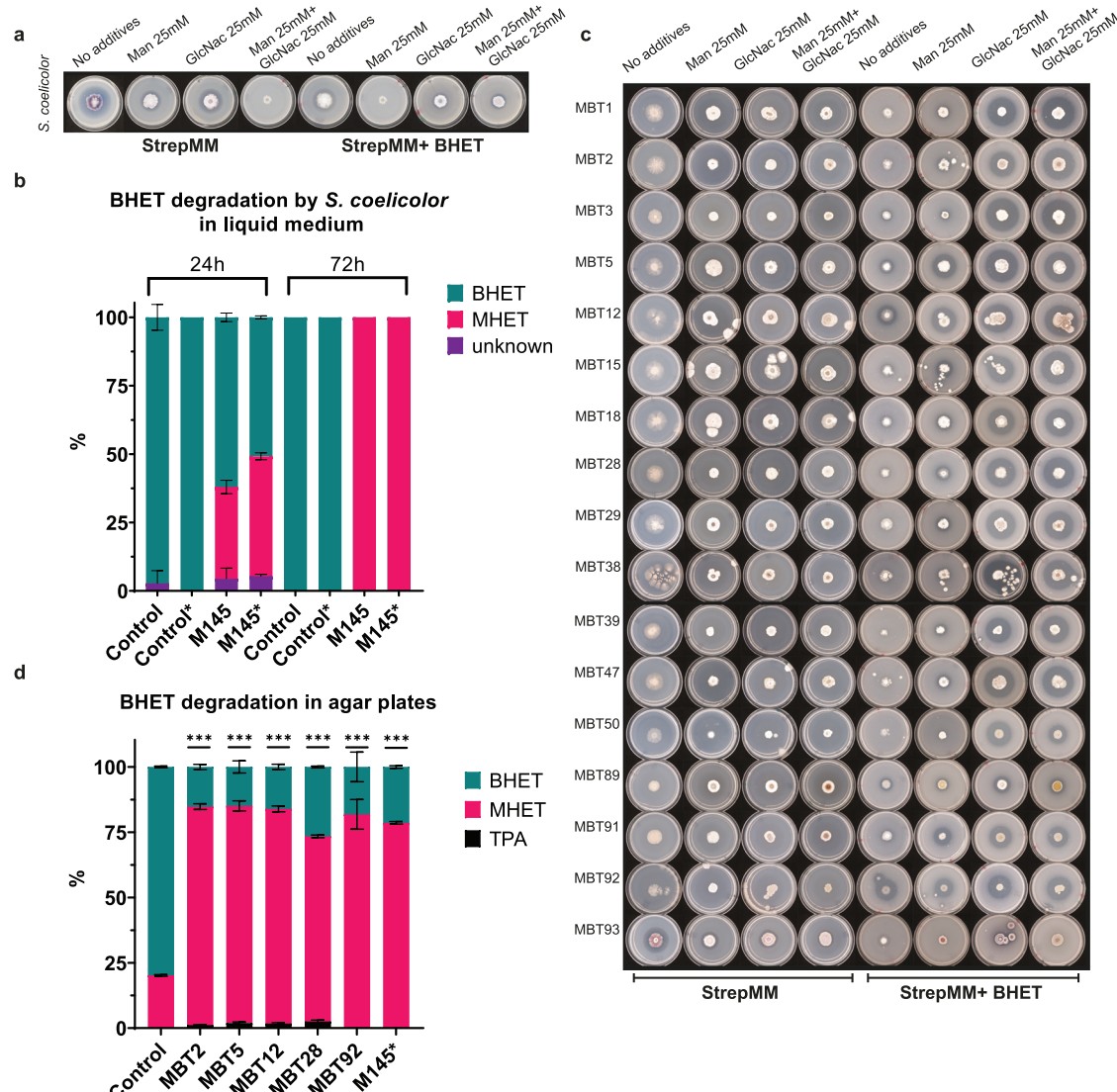

**Fig. 2 | The ability of *S. coelicolor* M145 and other Streptomycetes to degrade BHET. a** Degradation of BHET by *S. coelicolor* after 18 days of growth on StrepMM Difco agar with and without Mannitol, BHET, and GlcNAc. **b** The degradation of BHET by S. coelicolor in liquid medium (*n* = 3). The control samples are displayed as control and only contain NMM with BHET [10 mM]. The addition of GlcNAc [25 mM] to the cultures is indicated with an asterisk. The area percentages were calculated using GraphPad. The areas are presented in percentage compounds present in the culture. BHET is indicated in turquoise and MHET is indicated in magenta. Some impurities are present as a peak around 9.5 min

retention time, this compound is called unknown and is presented in purple. **c** Individual screen of all active strains on StrepMM Difco agar with and without mannitol, BHET, and GlcNAc after 10 days of growth. **d** Analysis of BHET degradation in agar plugs after 15 days of growth using LC-MS. The area percentage was calculated using GraphPad. *The agar plug of M145 was taken after 18 days. BHET is indicated in turquoise and MHET is indicated in magenta, TPA is represented in black. The statistical significance of the BHET degradation compared to the control is indicated with three asterisks. All error bars display the standard deviation, *n* = 3, and ***$p < 0,0001$. Individual data points are shown in Supplementary Fig. 4.

Thus, *S. coelicolor* M145 was grown in the presence and absence of BHET [10 mM], mannitol [25 mM] and N-acetyl glucosamine [25 mM] in all possible combinations. Indeed, a clearance halo was observed at 18 days of growth (Fig. 2a). Notably, the addition of GlcNAc as an inducer appeared to induce BHET degradation as well as enhance the development of the colony. Additionally, *S. coelicolor* is known to produce agarase[38], which resulted in 'sinking' halos as observed in the controls without BHET (Fig. 2a, StrepMM). These 'sinking halos' looked similar to clearance halos. Therefore, BHET degradation by *S. coelicolor* was confirmed by growth in liquid culture followed by Liquid Chromatography-Mass Spectrometry (LC-MS) (Fig. 2b). Strain M145 was precultured and inoculated in a minimal liquid medium without polyethylene glycol (NMM), with and without BHET and GlcNAc. Samples were taken after 24 h and 72 h and analyzed using LC-MS. When grown in NMM medium containing BHET, 35% of the BHET was

degraded within 24 h whereas 50% of BHET was degraded in the cultures containing GlcNAc. Indeed, the addition of GlcNAc significantly enhanced the degradation of BHET after 24 h (Fig. 2b, Supplementary data 1). After 72 h, all BHET was converted to MHET. With *S. coelicolor* M145 exhibiting a clear but modest BHET-degrading activity on plates, we set out to screen a diverse collection of 96 Actinobacteria of our strain collection[31] to investigate the spread of this characteristic and identify additional BHET-degrading bacteria. 36 strains were randomly chosen from this collection to perform an initial BHET toxicity test. Growth of all strains appeared to be somewhat delayed with 10 mM BHET, and impaired with 20 mM BHET and higher concentrations. Hence, a concentration of 10 mM of BHET was chosen for screening, with all tested strains able to grow and with several strains halos readily observed (Supplementary Fig. 1, Supplementary Table 1). A bulk screen for BHET degradation was performed with all 96 strains split over two 96-well

plates to provide them with enough space for development. Seven strains did not grow on the plates and were taken out of the screen (Supplementary Fig. 1, Supplementary Table 1). *Actinobacteria*, especially *Streptomyces*, depend on environmental cues and interspecies interactions for the induction of enzymes and secondary metabolites[39]. Consequently, it was important to investigate their individual BHET-degrading activity on separate plates. All strains showing halos in the bulk BHET screen were grown on individual plates (Supplementary Fig. 2, Supplementary Table 2). Of 38 active strains, only 17 strains exhibited individual activity on BHET after 10 days of growth (Fig. 2c, Supplementary Fig. 2). Interestingly, most strains required the addition of GlcNAc to express the BHET-degrading enzyme. The addition of GlcNac resulted in enhanced colony growth and development which likely also affected the degradation pattern of BHET. Only 5 strains, MBT5, 12, 15, 91, and 92 degraded BHET in the absence of GlcNAc. MBT92 appeared to exhibit constitutive expression showing a similar-sized halo in all conditions. To substantiate a good correlation between halo size and biochemical degradation, strains MBT2, 5, 12, 28, 92, and *S. coelicolor* M145 were incubated with approximately $2 \times 10^6$ spores on StrepMM Difco plates containing BHET and GlcNAc. Samples of agar were taken after 15 days using an agar excision tool to cut out part of the halo. Since *S. coelicolor* M145 displayed no activity after 15 days, halos were excised after 18 days. Samples were spun down using nylon spin columns, separating all liquids from the agar, and prepped for LC-MS analysis (Fig. 2d). In all strains, clear and significant conversion from BHET to MHET was observed. The MBT2, 5, 12, and 28 show traces of TPA indicating further conversion of MHET. A two-way ANOVA was performed to confirm the significance of the data, the results were displayed in Supplementary Data 1. LC-MS reference spectra can be found in Supplementary Fig. 3.

## Identification of LipA in BHET-degrading strains

To investigate if *lipA* is present in the strains showing activity in the individual screens displayed in Fig. 2c, PCR with degenerative primers was performed and the products were sequenced (Supplementary Fig. 5, Supplementary Note 1). Sequence alignment demonstrated substantial divergence between the signal sequences predicted with SignalP 6.0, while only minor variations in the overall enzyme coding regions were observed (Supplementary Fig. 6). The amino acid alignment showed variations between the proteins especially surrounding the catalytic triad and the C-terminal part of the protein (Fig. 3a). The enzyme sequences clustered into three variants: *S. coelicolor* (*Sc*LipA), MBT92 (*S92*LipA) only present in MBT92, and a variant present in all other active strains (*S2*LipA). Alphafold 2 enzyme structure prediction indicated small changes in enzyme structure with minor effects on the overall structure (Fig. 3b–d)[33].

## Scanning electron microscopy imaging of *Streptomyces* species on amorphous PET film

MBT2, 5, 12, 28, and 92 were grown with PET film in NMM with and without GlcNAc for 14 days. All strains form a biofilm surrounding the edges of the plastic films, thereby fixing them to the 12 wells plates. Samples were fixated, sputter coated, and visualized with scanning electron microscopy (SEM). All strains appeared to adhere well to the plastic films, forming hyphal structures colonizing the film (Supplementary Fig. 7). Some areas were heavily overgrown and did not allow for visualization of the film. Less cultivated areas did not show more damage than the control film, suggesting that no degradation of the PET films was achieved.

## Knock-out and overexpression of the three *lipA* variants in *S. coelicolor*

To obtain a better understanding of the role and function of LipA in vivo, a *lipA* knock-out mutant was constructed in *S. coelicolor* M145 ([40,41], Supplementary Note 2). Additionally, strains were constructed expressing one of the three different *lipA* variants under the control of the semi-constitutive pGAP promotor that is induced by simple sugar molecules[42,43]. To avoid

interference with the native *Sc*LipA and other regulatory elements, all overexpression constructs were cloned into strain Δ*lipA*. This resulted in strain S3 expressing *SclipA*, S5 expressing *S2lipA* and S7 expressing *S92lipA*. Two expression media were chosen namely NMM as a minimal medium with a defined carbon source and tryptic soy broth sucrose medium (TSBS). Expression was validated on both media using SDS-PAGE and western blot (Fig. 4a, b, d, e). Both NMM with 5% (w/v) glucose and TSBS yielded clear protein bands on western blot indicating that enzymes were successfully expressed.

BHET degradation was examined in vivo by inoculating $1.0 \times 10^8$ spores in 50 ml of NMM with glucose [5% (v/v)] and BHET or TSBS containing BHET. Samples were taken at 24, 48, and 72 h and analyzed using LC-MS. The percentages BHET, MHET, and TPA were calculated over the total peak area. After 24 h, while the spores are still germinating, limited activity is observed. After 48 h, on NMM, the knock-out shows significantly less activity on BHET than the wild-type strain. After 72 h, a clear difference was observed in the degradation pattern of the knock-out compared to the wild-type strain. On minimal medium, the overexpression constructs show 40–50% degradation, which is similar to the wild-type strain, although it is significantly higher than in the knock-out strain (parental strain) (Fig. 4c). Some residual activity, not related to LipA, was still observed in the knock-out strain. After 48 h on TSBS, no difference was observed in BHET-degrading ability of the wild-type strain compared to the knock-out strain. After 72 h, the wild-type strain degraded significantly more BHET than the knock-out strain. Unlike a minimal medium, the strains containing the overexpressed *lipA* variants showed drastic BHET degradation on TSBS after 48 h, with some variation between the different enzyme variants (Fig. 4f). After 72 h, the knock-out converted approximately 23% of the BHET to MHET, whereas the wild-type strain converted approximately 32%, the overexpression strains degraded all BHET after 72 h and TPA could be observed in the medium. The knock-out still showed some activity suggesting the presence of at least one other enzyme with low BHET-degrading activity. A two-way ANOVA clearly indicated the statistical significance of the results (Supplementary Data 2).

To observe physical changes within the cultures, automated time-lapse microscopy imaging was used to follow and visualize the development of the strains and degradation of the BHET. This microscope can make brightfield images at time intervals resulting in a time-lapse video (Supplementary Movies 1–6). It takes all strains around 10 h to start developing from spores (Fig. 5). Initially, no or only slight changes were observed in the presence and shape of the BHET crystals but over time the BHET crystals roughened and disappeared. The overexpression strains degraded the BHET crystals between 15 and 20 h, where the edges of the crystals started to roughen until the entire crystal fell apart (Fig. 5). The knock-out strain also degraded crystals, but complete degradation was only observed after 25 h. The wild-type strain degraded the BHET in approximately 23 h.

## Expression and in vitro characterization of LipA variants

For comprehensive in vitro biochemical analysis of the enzymes, the corresponding gene sequences were trimmed to remove their signal peptides, codon-optimized, synthesized, and cloned into pET16b (adding an N-terminal His-tag to the protein), and expressed in *E. coli* BL21 A-I. As positive controls, the genes encoding the PET40, *Tf*Cut2, the LCC, *Is*PETase, and PET46 were synthesized and expressed in the same way. All above-mentioned enzymes exhibit BHET and/or PET-degrading activity, however, their substrate preference, efficiency, optimal conditions, and stability are different. *Is*PETase has PET as its preferred substrate whereas PET40 is a promiscuous esterase which has a broad substrate range. *Tf*Cut2 and LCC are cutinases that display high activity on cutinase-like substrates such as the para-nitrophenyl substrates and lipase-like substrates as tributyrin[25,26,44–47]. PET46 was annotated as feruloyl esterase which should be more specific for the degradation of feruloyl polysaccharides and was shown to be more efficient in MHET degradation[48]. Expression and purification of the corresponding enzymes were confirmed on SDS-PAGE (Fig. 6a) and western blot (Fig. 6b) with protein bands around 32 kDa. His-tag purification

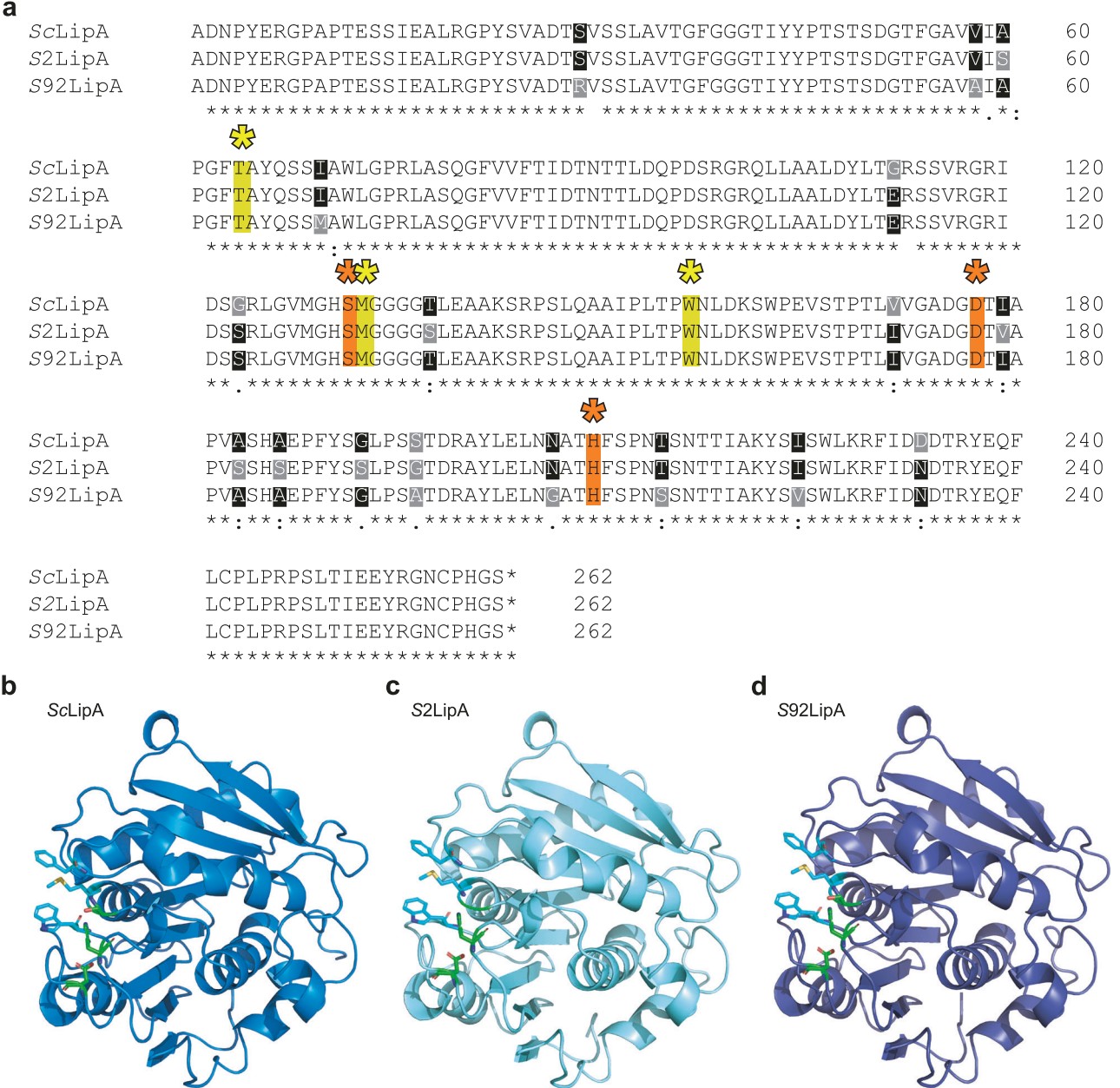

**Fig. 3 | Comparison of the Lipase A variants. a** Sequence comparison of the *Sc*LipA, *S2*LipA, and *S92*LipA indicating the binding residues with yellow asterisks/highlight and the catalytic triad with orange asterisks/highlight. Sequence differences from the different lipases are highlighted in gray, when only one sequence differs, the conserved amino acids are highlighted in black. **b–d** Predicted model of the structure of *Sc*LipA (**b**), *S2*LipA (**c**) and *S92*LipA (**d**) constructed with AlphaFold 2. The catalytic triad residues are displayed in green and the binding domain residues are displayed in cyan.

resulted in the LipA samples displaying two bands on SDS-PAGE. The higher band (~60 kDa) was always observed on SDS-PAGE and inconsistently observed on western blot; Fig. 4e is an example where the higher band is clearly observed whereas in Fig. 6b this band is not observed. We hypothesize that this higher band corresponds with a dimeric form where the His-tag is poorly accessible. Native zymogram analysis clearly showed that enzyme activity only resided with the lower band for the Lipase A variants (Fig. 6c). PET40, *Tf*Cut2 and the LCC did not migrate through the native gel, possibly due to a substantial difference in isoelectric point.

The optimal pH and temperature for enzyme activity were determined using para-nitrophenyl dodecanoate as substrate according to the method of Altammar and colleagues[49]. Within a pH range from pH 3 to pH 8, for one hour at 30 °C, pH 7.0 appeared as the optimal pH. A similar approach was taken to determine the optimal reaction temperature, which was 25 °C (Fig. 7a–c).

Using these conditions a colorimetric assay was developed to examine BHET degradation of the LipA variants and all above-mentioned controls. 2.0 µg/ml of enzyme was incubated with 0.5 mM of BHET for 0, 24, 48 h (Fig. 7d). *S2*LipA and *S92*LipA degraded around 0.2 mM BHET in 24 h increasing to 0.25 mM in 48 h. *Sc*LipA showed slightly less degradation activity compared to the other variants up to 0.15 mM in 24 h. PET46 showed similar activity after 24 h as *S2*LipA but increased towards 0.3 mM after 48 h. The PET40, *Tf*Cut2, and LCC degraded approximately 30 mM of BHET within 24 h increasing to 40 mM in 48 h. The *Is*PETase showed the highest activity converting all substrates within 24 h. Numerical data and Statistical analysis are provided in Supplementary Data 3.

To obtain a preliminary view of the polymer activity of the Lipase A enzymes, amorphous PET films were inoculated at pH 7.0 and 25 °C with 15 µg/ml of enzyme for 7 days and visualized using scanning electron

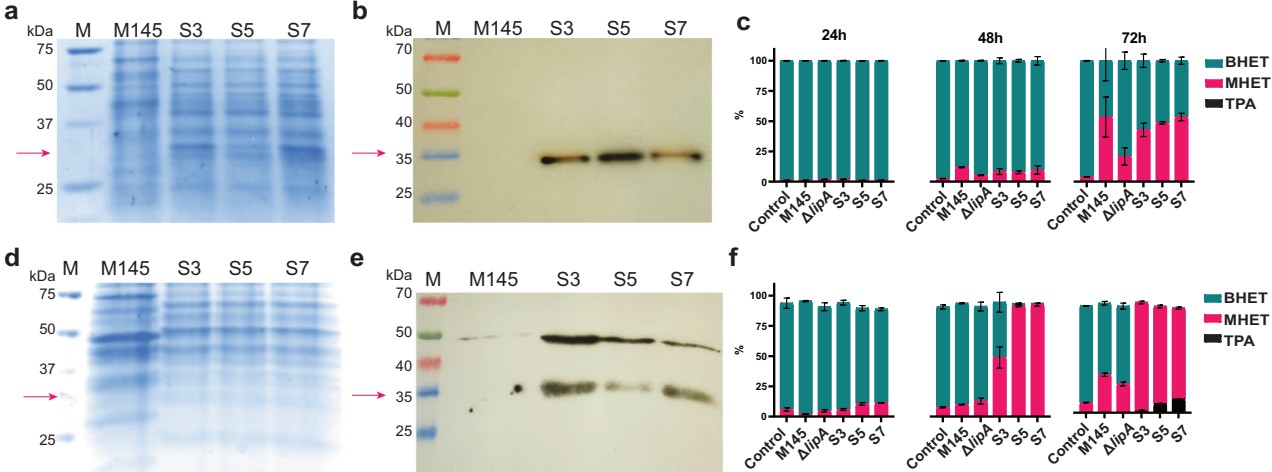

**Fig. 4 | Expression of the LipA variants in *S. coelicolor* M145 ΔlipA in NMM and TSBS medium. a** SDS-PAGE of concentrated samples on NMM medium, faint band around ~32 kDa in the overexpression strains, this band is not present in wild-type strain M145. **b** Western blot of NMM samples showing clear signal around ~32 kDa. **c** Analysis of BHET degradation in NMM of the wild-type strain, ΔlipA, S3, S5, and S7 using LC-MS (*n* = 3). Samples were taken at 24 h, 48 h and 72 h. The percentage of BHET is presented in turquoise, the percentage of MHET in magenta, and the percentage of TPA in black. The area percentage was calculated using GraphPad. **d** SDS-PAGE of concentrated samples on TSBS medium, a faint band around ~32 kDa was observed in the overexpression strains this band was not present in wild-type strain M145. **e** Western blot of TSBS samples showing a clear signal around 32 kDa. **f** Analysis of BHET degradation in TSBS of the wild-type strain, ΔlipA, S3, S5 and S7 using LC-MS. Samples were taken at 24 h, 48 h and 72 h. The percentage of BHET is presented in turquoise, the percentage of MHET in magenta, and the percentage of TPA in black. The error bars display the standard deviation. All original gel pictures are shown in Supplementary Fig. 8. The graphs containing all individual data points are shown in Supplementary Fig. 9.

microscopy (SEM). Since the activity of the *Is*PETase exceeded the activity of all other enzymes it was used as a positive control. Since no significant difference was observed between PET40, *Tf*Cut2, LCC, and PET46 after 48 h, only *Tf*Cut2 was taken as an additional positive control. Samples only incubated with buffer show a smooth clean surface. PET films incubated with *Is*PETase show clear damage in the form of dents and cavities (Fig. 8). *Tf*Cut2 showed roughening of the surface and at 10,000× magnification small indents. *S2*LipA showed a similar pattern as *Tf*Cut2 although the top layer of the plastic appeared not to be completely degraded. Both *S92*LipA and *Sc*LipA showed limited to no activity which is in line with the enzyme assay (Fig. 8).

## Discussion

Understanding and harnessing nature's response to abundant plastic pollution may help find new solutions for sustainable plastic depolymerization and recycling. However, and importantly, the abundance of microorganisms that have adapted to the utilization and degradation of plastics is not known, and the evolution and conservation of genes and proteins involved remains largely unexplored.

In this research, we have investigated the BHET/PET-degrading activity of a PHE previously annotated as Lipase A (LipA) from *Streptomyces coelicolor*. Lipase A was originally identified via whole genome studies, never further investigated for its activity, and automatically annotated as lipase. In a bulk screen of 96 *Streptomyces* strains from various origins, 44% was able to degrade BHET in at least one of the tested conditions. When tested individually, 18% of the strains showed clear degrading activity. Since in the bulk screen neighboring strains showed BHET-degrading activity which was not confirmed for all strains when tested individually, interaction between strains may have an important effect on the expression of BHET-degrading enzymes. In addition, GlcNAc appeared to induce BHET-degrading activity. This is likely to be related to GlcNAc-mediated enhanced development of the *Streptomyces* strains under famine conditions[31,39]. So far, limited information is available on the regulation and induction of PHEs. Unveiling and understanding the conditions in which PHEs are expressed will shine a light on environmental cues that may affect the emergence of biodegradation of polyesters in nature.

*LipA* was present in 15 different *Streptomyces* strains as a highly conserved gene with some variations. Three variants, *Sc*lipA (present in *S. coelicolor*), *S92*lipA (present in MBT92), and *S2*lipA (present all other active strains) were further investigated. This situation is comparable to the presence of PET40 in several *Amycolatopsis* and other *Streptomyces* species[28]. However, in this study, the BHET-degrading activity in the original host strains was not investigated. We chose to investigate the in vivo activity of *Sc*LipA by comparing a knock-out strain with the wild-type strain, in both rich and minimal medium; the knock-out exhibited a significant decrease in its BHET-degrading activity. Since some remaining BHET degradation was still observed in the knock-out strain, at least one other enzyme is likely to be involved in BHET degradation. However, this activity most likely resides with another type of enzyme since no other homolog was found during the homology search. All three LipA variants exhibited significantly enhanced degradation of BHET when expressed in the knock-out strain under a semi-constitutive promoter, on both rich and minimal medium.

All enzyme variants were found to have an optimal pH of 7 and an optimal temperature of 25 °C. The enzymes were active in a range from pH 5-10, similar to the PET40[28]. However, in contrast with the PET40, the LipA variants appeared to lose most activity above 40 °C. Colorimetric BHET assays confirmed the BHET-degrading activity of *S2*LipA and *S92*LipA under optimal conditions, however, all positive controls displayed higher activities consistent with the literature. The relatively rapid loss of activity of the LipA enzymes is likely related to the stability of the enzymes.

Incubation of amorphous PET film with purified *S2*LipA caused the formation of indents and roughening of the surface, similar to the observed activity of PET-degrading cutinase *Tf*Cut2 in suboptimal conditions[26]. Thus, the *S2*LipA variant showed some degrading activity on PET film, in contrast to the other two variants. Since the activity on amorphous PET is low, we expect that *S2*LipA will not show activity on post-consumer PET. To obtain high activity on polymeric PET further enzyme engineering and optimization will be required.

While the structure and catalytic activity of the LipA variants are more similar to PETases than to BHETases[50], our enzymes seem to exhibit more activity on BHET than on amorphous PET. Indeed, while PET is not the main substrate, PET and BHET degradation appear as promiscuous activity, similar to the moonlighting effect described for the LCC, *Tf*Cut2, and PET40[19–26]. Furthermore, the activity and behavior of the LipA enzymes are not in line with the original annotation as lipase. The observed activity is

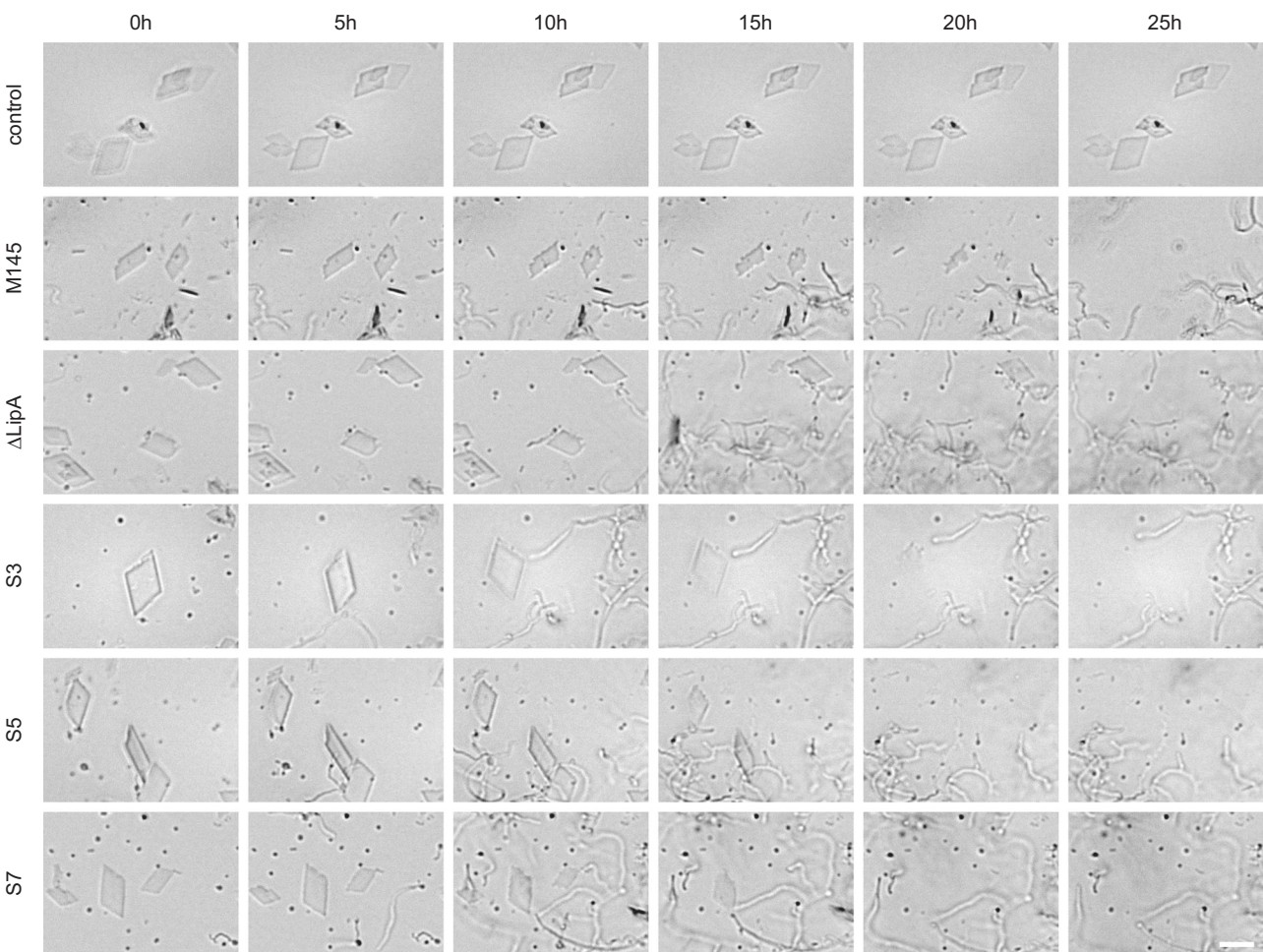

**Fig. 5 | Time-lapse images of BHET degradation by *S. coelicolor* M145, ΔlipA, S3, S5, and S7.** Time-lapse images at 5-h intervals of wild-type and ΔlipA mutant and overexpression strains S3, S5, and S7 in the presence of BHET particles (diamond-shaped particles). A negative control lacking bacterial inoculation was used to demonstrate that the BHET particles do not undergo natural degradation over time (indicated as control). The scale bar in the top right corner is 10 μm.

more consistent with PETase-like esterase and/or feruloyl-esterase activity comparable to PET40. Since the main activity was observed on BHET we suggest re-evaluating the annotation of those genes and enzymes and changing it to BHET-degrading esterase A, or BdeA. *S92*LipA and *Sc*LipA showed lower BHET-degrading activity than *S2*LipA which most probably is due to structural variations in the enzymes caused by the various amino acid substitutions. In comparison to *S92*LipA and *Sc*LipA, *S2*LipA contains a region close to important catalytic residues and substrate binding residues where three alanines have been substituted by serines (Fig. 3a). This may enable enhanced formation of hydrogen bonds with the substrate and/or enhanced enzyme stability[51–53]. Since all of the strains in our study are environmental isolates, we hypothesize that the abundance of the *S2*LipA variant in these strains is caused by evolution towards enhanced activity on polar substrates and a more stable esterase and not toward plastic degradation per se. Yet, current pollution rates may push the environmental evolution of those enzymes towards enhanced BHET/PET affinity. The current abundance of BHET-degrading enzymes in nature indicates that nature can indeed take on the challenge of degradation and bioremediation. By further investigating and understanding this phenomenon, the microbial and enzyme characteristics involved, and applying those features, we may move towards mitigating plastic pollution for a more sustainable future.

## Materials and methods
### Strains and culturing conditions
For the initial tests, *Streptomyces coelicolor* M145[17] and Actinobacteria from the MBT strain collection of the Institute of Biology Leiden were used for screening. These strains were isolated from the Himalayan mountains, Qinling mountains, Cheverny and Grenouillere France, and the Netherlands[41]. The *Streptomyces* plates were incubated at 30 °C for 10 days or longer before investigating the BHET degradation. *E. coli* was grown on Luria-Burtani (LB) agar or in LB cultures overnight at 37 °C.

Routine *Streptomyces* manipulation, growth, and preparation of spore stocks was performed according to the *Streptomyces* manual[54]. Spore suspensions were obtained by growing the strains on Soy Flour Mannitol agar (SFM[55]) until sporulation.

### Homology search
Potential homologs of LCC (Accession number G9BY57[44]), *Tf*Cut2 (Accession number Q6A0I4[25]), PET40 (Accession number WAU86704.1[28]) and *Is*PETase (Accession number A0A0K8P6T7[29]) in the Actinobacterial genomes were identified by performing a BLASTp for the genome of *Streptomyces coelicolor* (NCBI 100226)[30]. BLASTp was run in the default setting. The hits were ranked based on their BLAST score, their query cover, their percent identity, and the presence of the catalytic triad and substrate binding site.

### Bulk screens
The strains are inoculated on the plates using the stamping method. A stamp fitting a 96-well plate was autoclaved, placed in the stamping device, and placed in the 96-well plate, containing 100 μl of each spore stock in separate wells, and stamped on the plates of interest. Each spot contained approximately 2 μl of spore solution. The plates were grown for 10 days at 30 °C. The spore plates were stored at −20 °C.

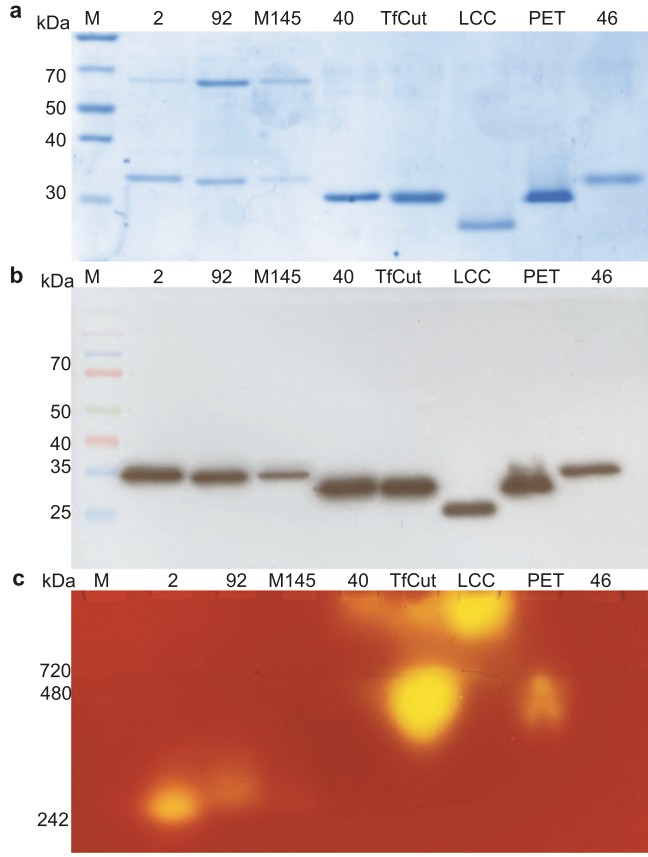

**Fig. 6 | Expression and purification of Lipase A enzymes in E. coli. a** SDS-PAGE of the purified Lipase A variants, along with the PET40 (40), *Tf*Cut2, LCC, *Is*PETase (PET) and PET46 (46); 1: Ladder, multicolor broad range protein ladder; 2: *S2*LipA sample (30 kDa); 3: *S92*LipA (30 kDa); 4: *Sc*LipA (30 kDa), 5: PET40 (28 kDa), 6: *Tf*Cut2 (28 kDa), 7: LCC (27 kDa), *Is*PETase (27 kDa), 8: PET46 (30 kDa). The Lipase variants show two bands one around 60 kDa and one around 35 kDa. **b** Western blot using a His-antibody was conducted to identify the purified enzymes of the Lipase A variants. 1: ladder, Spectra multicolor broad range protein marker; 2 till 8: same samples as for the SDS-PAGE **c** Zymogram of 750 ng of purified enzymes and LipA variants on 1% tributyrin. The Lipase A variants, PET40, *Tf*Cut2, LCC, and PETase show degradation of tributyrin (yellow spots). NativeMarker used as a marker and visualized by overlaying Native SDS-PAGE with a zymogram. All original gel pictures are shown in Supplementary Fig. 10.

## Plate assays

Plate assays were performed using minimal medium plates with different types of agars. Upon media selection, *Streptomyces* minimal medium (StrepMM[54]) Difco agar was chosen as the standard condition[56]. Plates and additives are described in Supplementary Table 3. Bis(2-hydroxyethyl) terephthalate (BHET) provided by Sigma (Cas: 959-26-2, PN 465151), N-acetyl glucosamine (GlcNAc) provided by Sigma (Cas 7512-17-6, PN A4106).

## Individual screens

For the individual screens, 2 μl of uncorrected spore solution was spotted on 5 cm agar plates, and grown at 30 °C for 10 days.

## Agar excision agar samples

*S. coelicolor* M145, and *Streptomyces* species MBT2, MBT5, MBT12, MBT38, and MBT92 with $2 \times 10^6$ spores were inoculated on StrepMM Difco with GlcNAc [25 mM] and BHET [10 mM]. After 14 days of growth, a part of the halo was excised using an agar excision tool. The agar and liquids were separated using clear spin-filtered microtubes (Sorenson BioSciences) spun at 10,000 rpm for 15 min. The flow-through was stored at −20 °C and prepared for LC-MS analysis according to the sample preparation protocol.

## Identification variants *Streptomyces* strains

For the identification of the Lipase A variants in the *Actinobacteria* collection genomic DNA of the active strains was isolated. A PCR was performed using degenerative primers (Supplementary Table 4) based on the sequence of the *2lipA* gene. PCR was performed with Phusion according to the Phusion protocol provided by Thermo Fischer Scientific (F531S). The blunt-end PCR products were cloned in pJET2.1 using the CloneJET PCR cloning kit from Thermo Fischer Scientific (K1231) and sequenced by Macrogen using Sanger sequencing.

## CRISPR/Cas knock-out

To obtain a LipA knock-out strain, the CRISPR-Cas9-based pCRISPomyces was used[40,41]. In this system, the CRISPR-Cas9-mediated double-stranded break will be repaired via homology-directed repair. Two homologous arms of approximately 1000 bp with a 40 nt overlap at both sides of *lipA* have been amplified via PCR on the genomic DNA of M145. These homologous arms can be digested into the pCRISPomyces-2 via a three-piece Gibson Assembly using the XbaI site. Additionally, a sgRNA will be annealed into the plasmid using Golden Gate cloning. Primers for these homologous arms bordering LipA and sgRNA were made using the protocol of Cobb and colleagues and are provided in Supplementary Table 5. Assembled plasmids were confirmed via sequencing. Assembled plasmids were transferred to *Streptomyces coelicolor* M145 via conjugation with *E. coli* ET12567 harboring the pUZ8002 plasmid[57,58]. Single colonies were streaked on SFM with apramycin [50 μg/ml] to check for true apramycin resistance. Resistant colonies were picked to TSBS for gDNA isolation. Correct gene knockouts were confirmed via diagnostic PCR and sequencing Supplementary Fig. 11. Plasmid loss was achieved by restreaking strains for several generations on SFM agar without antibiotic pressure at 37 °C and checking for the loss of apramycin resistance[49,50].

## Construction overexpression strains

Overexpression strains were obtained by using the pSET152 integrative plasmid[43]. To create overexpression strains for LipA and its variants, the genes were transferred to a pSET152 integrative plasmid. Genomic DNA of *S. coelicolor* M145, *Streptomyces* sp. MBT2 and *Streptomyces* sp. MBT92 was isolated and the *lipA* PCR amplified with overhangs containing an NdeI site on the 5′ and a His-tag followed by a BamHI site on the 3′ end of the product (Supplementary Table 6). pSET152 was linearized with BamHI and NdeI and PCR amplified *lipA* genes were digested into the plasmid using T4 ligase. Assembled plasmids were confirmed via restriction analysis with BamHI and NdeI and sequencing (Supplementary Fig. 12). *Sc*LipA and its variations were cloned into pSET152 under the regulation of strong semi-constitutive promotor pGAP[42]. pSET152 was transferred to *Streptomyces coelicolor* M145 via conjugation as described in refs. 57,58. Single colonies were streaked on SFM with apramycin to select for the correct integration of the plasmid. All strains obtained are described in Supplementary Table 7.

## Liquid cultures for enzyme expression

For liquid chromatography-mass spectrometry (LC-MS) analysis, $1.0 \times 10^8$ spores were inoculated in 50 ml liquid minimal medium without PEG (NMM[54]) + 5% glucose (w/v) and enzyme expression medium Tryptic soy broth sucrose (TSBS[54]) containing 10 mM BHET. Samples of 1 ml were taken after 24, 48, and 72 h.

## Liquid chromatography-mass spectrometry (LC-MS)

Upon thawing, cell-free media was prepared for the LC-MS run by dilution in 1:1 in acetonitrile at a final volume of 1 ml and passed through 0.2 μm filters (Sartorius). Analysis of TPA, MHET, and BHET. LC-MS analyses were performed on a Shimadzu LC-20AD system with a Shimadzu Shim-Pack GIST-HP C18-AQ column (3.0 × 150 mm, 3 μm) at 40 °C and equipped with a UV detector monitoring at 240 and 260 nm. The following

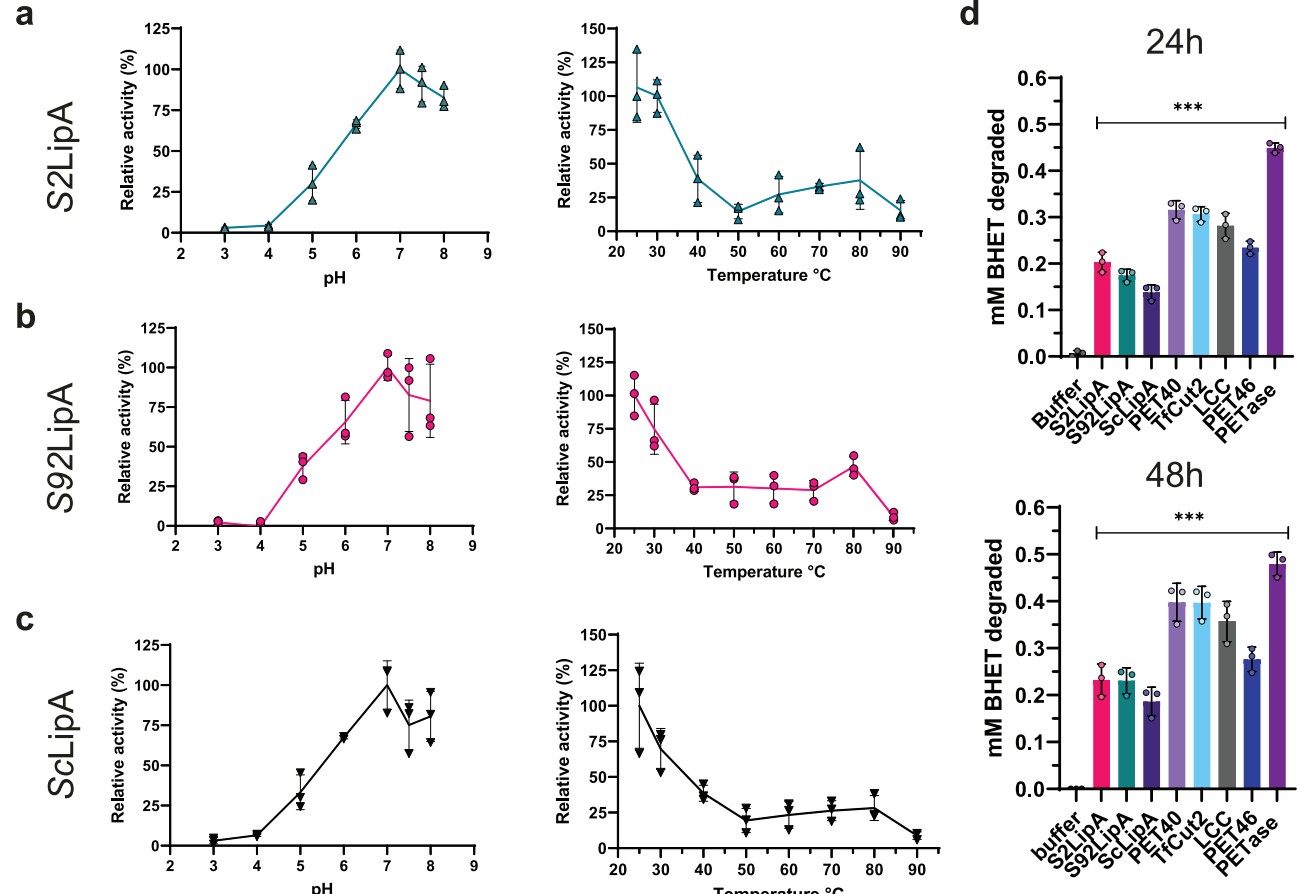

**Fig. 7 | Determination of optimal enzyme conditions and BHET degradation under optimum conditions. a–c** Relative activity on para-nitrophenol dodecanoate at different pH for $S2$LipA (**a**), $S92$LipA (**b**) and $Sc$LipA (**c**) ($n = 3$). The graphs are overlays of the graph containing the error bars and the same graph with the individual data points original graphs are displayed as Supplementary Fig. 13. **d** Enzymatic BHET degradation using colorimetric assay after 24 and 48 h. The activity of buffer control (black) as a negative control, $S2$LipA (magenta) $S92$LipA (turquoise), and $Sc$LipA (dark purple), the positive controls are displayed as PET40 (lavender), $Tf$Cut2 (light blue) LCC (black), PET46 (dark blue), and $Is$PETase (purple). The activity was displayed as the concentration of BHET degraded in mM. The indicated significance displays the significant difference with the buffer ($n = 3$). The error bars display the standard deviation, ***$p < 0,0001$.

solvent system, at a flow rate of 0.5 ml/min, was used: solvent A, 0.1% formic acid in water; solvent B, acetonitrile. Gradient elution was as follows: 80:20 (A/B) for 1 min, 80:20 to 45:55 (A/B) over 6 min, 45:55 to 0:100 (A/B) over 1 min, 0:100 (A/B) for 2 min, then reversion back to 80:20 (A/B) over 1 min and 80:20 (A/B) for 2 min. This system was connected to a Shimadzu 8040 triple quadrupole mass spectrometer (ESI ionization).

The reference chromatogram used for the identification of BHET and its monomers [~1 mg/ml each]. The area percentage was calculated using GraphPad Prism. The statistical analysis of all LC-MS data consists of a two-way ANOVA using the default setting of GraphPad Prism ($p < 0.05$, $n = 3$). A multiple comparison analysis was performed providing insights into the significance of each compound present in each sample. The outcomes of the statistical analysis are provided in Supplementary Data 1–3 respectively.

## Microscopy

The Lionheart FX automated microscope (BioTek) was used for the time-lapse imaging of BHET particle degradation. In total, $3.5 \times 10^4$ spores were precultured in TSBS + nalidixic acid [50 μg/ml] for 48 h. Spores were precultured in TSBS containing nalidixic acid [50 μg/ml] for 48 h. Of this solution, 5 μl was inoculated in 200 μl TSBS containing ampicillin [50 μg/ml] and BHET [0.25 mM]. This solution was added to a Greiner Bio-One SensoPlate 96-well, non-treated black plate with a clear bottom and spun down to settle the mixture. Bright-field pictures were taken every 15 min over the course of 60 h on a 40× magnification at 30 °C.

## Expression in *E. coli*

$S2$LipA, $S92$LipA, $Sc$LipA, LCC, $Tf$Cut2, PET40, and $Is$PETase without signal sequences were codon optimized for *E. coli* and ordered with GeneART from ThermoFisher Scientific (Netherlands). The genes arrived in the Thermofisher plasmid pMA containing an ampicillin resistance. PET46 was provided by the Streit Lab of the University of Hamburg and amplified using primers with BamHI and NdeI overhangs (Supplementary Table 8) All codon-optimized DNA sequences are provided in Supplementary Note 3.

The plasmids were transformed into chemically competent *E. coli DH5α* cells for amplification and isolated using the "GeneJET plasmid miniprep Kit" (Thermofischer, K0503). After plasmid isolation, the *lipA* variants were cloned into the pET16b plasmid using restriction/ligation with restriction enzymes BamHI and NdeI [10 U/μl] (Thermo Fischer Scientific). BamHI was added half an hour after incubation with NdeI to achieve higher efficiency. Following restriction, the restriction enzymes were inactivated by incubating the samples for 20 min at 80 °C in a water bath. The *lipA* genes and the linearized pET16b plasmids were extracted from the gel using the "GeneJET GEL extraction Kit" (Thermofischer, K0692). T4 ligase was used according to the user manual of the manufacturer. See Supplementary Table 9 for an overview of the used plasmids for protein expression.

Transformed *E. coli BL21 A-I (*Invitrogen C607003*)* strains were grown on agar plates with ampicillin [100 μg/mL] at 37 °C or in LB liquid medium at 37 °C at 200 rpm (Supplementary Table 10).

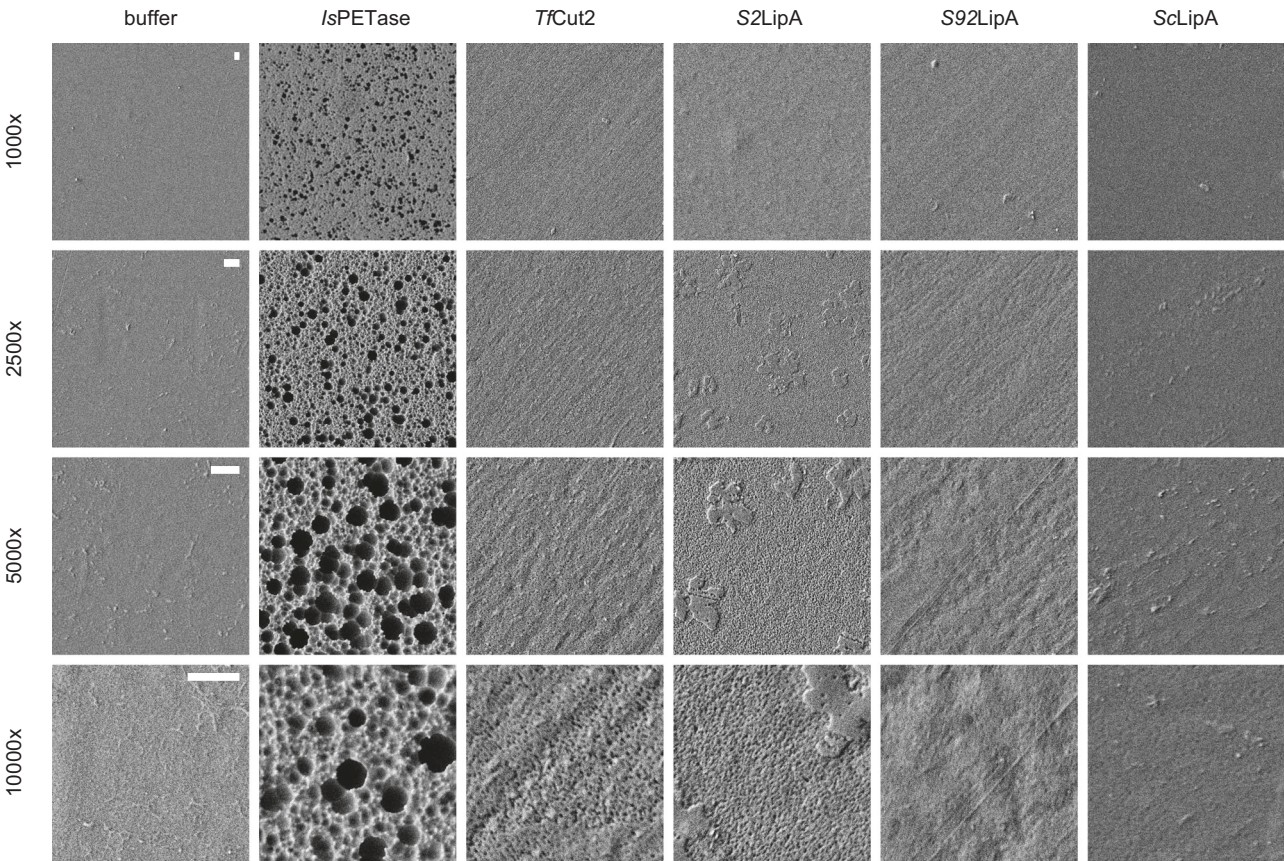

**Fig. 8 | Effect of *Is*PETase, *Tf*Cut2, *S2*LipA, *S92*LipA, and *Sc*LipA on amorphous PET film.** Amorphous PET films after 7 days of incubation with 15 µg/ml enzyme at 25 °C pH 7 at 1000–10.000× magnification. The scalebars in the left panels are equal to 1 µm and apply to all images of the same magnification. Each row displays a different magnification whereas the columns show a different enzyme of incubation.

## Enzyme purification

For protein production, 20 ml precultures with ampicillin (100 µg/ml) of the transformed *E. coli* BL21 A-I were made following the standard culture conditions. 5 ml of these precultures were transferred to 500 ml TB containing ampicillin. When an OD value between 0.6 and 0.8 was reached, the cultures were induced using arabinose [0.2% (w/v)] and IPTG [0.5 mM].

Pellet the 500 ml cultures were harvested by ultracentrifugation (Himac) for 30 min, 6000 rpm at 4 °C. The harvested pellet was transferred to 50 ml falcon tubes and frozen in liquid nitrogen. The frozen pellets were stored at −20 °C.

Next, the defrosted pellet was resuspended in 15 ml 50 mM Tris-HCl and 300 mM NaCl pH 7.5. These solutions were sonicated (Bandelin, Sonopuls) 3 times for 30 s at 15% Amplitude. Subsequently, to remove the insoluble faction's ultracentrifugation (Himac) took place for 60 min, 10,000 rpm at 4 °C. The Sample was filtered through a 0.2 µm filter (Filtropur S 0.2 µm, Sarstedt).

To obtain purified Lipase A variants, His – affinity purification using stationary columns system was performed, following standard His-tag purification protocols. During purification, the column was washed with 10 ml 50 mM Tris-HCl, 300 mM NaCl containing 75 mM imidazole pH 7.5, and eluted with the same buffer containing 500 mM imidazole. After purification, the fractions were desalted using buffer C desalting columns (Cytiva, PD-10) according to the protocol of Cytiva columns. The resulting purified Lipase A variants were stored at −20 °C in 25 mM Tris-HCl containing 150 mM NaCl and 40% glycerol.

## SDS-PAGE, Western blot, and Zymogram

For SDS-PAGE and Western blot, 50 ml of TSBS/NMM + 5% glucose (v/v) were inoculated with 50 µl dense spore prep and incubated for 24 h. Cultures were centrifuged at $4000 \times g$ for 10 min and supernatant was filtered through a 0.2 µm filter. The supernatant was concentrated using a Viaspin column (Sartorius 5 kDas). Protein concentrations were estimated and normalized to 200 ng/µl by performing a Bradford assay.

Overall, all SDS-PAGEs contained 12% acrylamide and were run for 20 min at 70 V to stack the proteins on the gels. Further, the gel was run at 150 V until the loading dye reached the bottom of the gel. SDS-PAGE gels were stained with Coomassie-blue staining. For the western blot, the gels were transferred using a BioRad Trans-blot Turbo and the corresponding transfer packs (1704157EDU) according to the mixed gel protocol of BioRad. Gel was washed using Tris-buffered saline (TBS) buffer and blocked using Tris-buffered saline with 0.5% Tween 20 (TBST) buffer containing 1% Elk milk. The blot was blocked between 60 and 90 min. His-antibody was added to a final concentration of 1 µg/ml and incubated overnight (K953-01). The blot was rinsed with water and washed 4 times with TBST, the blot was then incubated with luminol for 1 min, (product number) dried, and developed on X-ray film.

For the zymogram, an 8% native gel, was run for 30 min at 70 V and 2 h at 150 V. The gel was rinsed with demi water and washed with demi water and equilibrated by incubating 3 times for 30 min with 25 mM NaCl. After, the gel is placed on a 1% tributyrin, 0.5% agarose plate, and incubated at 30 °C for 1.5 h[59].

## Standard enzyme assays

The concentration of enzyme was estimated using the Bradford method (Bio-Rad, Bradford 1x Dye Reagent 5000205). The esterase/cutinase activity was tested using para-nitrophenyl dodecanoate (Sigma-Aldrich, 61716) The protocol followed was based on the protocol of Altammar and colleagues with minor adjustments[49]. For the optimal pH test, 50 mM citrate buffers

**Article**

ranging from pH 3 to pH 7 were used, and for pH 7.5 and 8 50 mM Tris-HCl buffer was used. The incubation step of 10 min was prolonged to 1 h. The reaction was terminated using 0.1 M sodium carbonate[49]

## Colorimetric assay BHET degradation

The colorimetric assay was performed according to the methods of Beech and colleagues[60]. 2 μg/ml of the enzyme was incubated with 0.5 mM of BHET for 0, 24, and 48 h and measured at 615 nm in the Tecan M Spark. A reference line was made by adding BHET in the concentrations of 0.5 mM to 0 mM in steps of 0.05 mM an excess of PET40 was added to convert all BHET to MHET.

The amount of BHET degraded was calculated with GraphPad using the above-mentioned reference. The statistical analysis consists of a one-way ANOVA using the default setting of GraphPad Prism ($p = 0.05$, $n = 3$). A comparison was made between the means of each enzyme treatment providing insights into the significance of the BHET-degrading activity of each enzyme. The outcomes of the statistical analysis are provided in Supplementary Data 3. The error bars in the figures represent the standard deviation.

## SEM

The effect of Streptomyces on amorphous plastic films was investigated by incubating $10^7$ spores in 3 ml of NMM with amorphous PET for two weeks at 30 °C. The samples were fixed with 1.5% glutaraldehyde (30 min). Subsequently, samples were dehydrated using a series of increasing ethanol percentages (70%, 80%, 90%, 96%, and 100%, each step 30 min) and critical point dried (Baltec CPD-030). Hereafter the samples were coated with 10 nm Platinum palladium using a sputter coater and directly imaged using a JEOL JSM6700F. When investigating enzymes on PET film, 15 μg/ml of the enzyme was incubated for 7 days, the films were washed with water and 70% ethanol, air dried, sputter coated with 10 nm Platinum palladium, and visualized using a JEOL JSM6700F.

## Statistical analysis and reproducibility

The statistical analysis of datasets consists of a one-way or two-way ANOVA using the default settings of GraphPad Prism ($*p < 0.05$, $**p < 0,01$, $***p < 0,0001$, $n = 3$ biological replicates). Multiple comparisons were made, and the displayed significance in the figures consists of the negative control compared to the samples. The outcomes of the statistical analysis and numerical data of the graphs are provided in Supplementary Data 1–3. The error bars in the figures represent the standard deviation.

## Reporting summary

Further information on research design is available in the Nature Portfolio Reporting Summary linked to this article.

## Data availability

The protein sequences used for the alignments in Fig. 1 and further expression can be found in the Uniprot database using the following accession numbers LCC (Accession number G9BY57[44]), TfCut2 (Accession number Q6A0I4[25]), IsPETase (Accession number A0A0K8P6T7[29]) and (ScLipA, accession number Q9L2J6[30]). The protein sequence of PET40 and (NCBI accession number WAU86704.1[28]), PET46 was obtained via the PAZy database (NCBI accession number RLI42440.1). The source data underlying Fig. 2 can be found in Supplementary Data 1. The DNA and protein sequences underlying the information in Fig. 3 are provided in Supplementary Note 1 and Supplementary Fig. 6. The original gels of Fig. 4 can be found in Supplementary Fig. 8, the source data of the graphs of Fig. 4 can be found in Supplementary Data 2 and the graphs containing all individual data points is provided as Supplementary Fig. 9. The supplementary movies supporting Fig. 5 can be found as Supplementary Movies 1–6. The original gels of Fig. 6 are provided as Supplementary Fig. 10. All source data for Fig. 7 is provided in Supplementary Data 3, and the non-overlayed graphs of Fig. 7 can be found in Supplementary Fig. 13. Any remaining data of this study are available upon request.

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

## Acknowledgements

We thank Lennart Schada von Borzyskowski for providing BL21 A-I and the pET16b. Additionally, Chaoxian Bai for providing the CRISPR system for *Streptomyces* and his advice on using it. Erik Vijgenboom and Jean Richard Quant for providing pSET152pGAP. The group of Wolfgang Streit from the University of Hamburg provided the plasmid containing PET46. Kees van den Hondel for this introduction into the lipase substrates. Marco Blasioli for his background work on LipA. Mia Urem for suggesting GlcNAc as a possible inducer. Somayah Elsayed for explaining the MZ-mine software.

Nathaniel Martin for his advice and support on setting up the LC-MS methods. Finally, Davy de Witt for preparing all the media.

## Author contributions

J.A.V.: conceptualization, data acquisition (including LC-MS, SEM, and Lionheart) data analysis, figure design, and writing. M.C.: Data acquisition (including LC-MS), data analysis, and writing; S.L. homology search, 3D-structure modeling, PyMOL analysis, and figure design; A.M.: Data acquisition; C.B.: Data acquisition; P.I.: development LC-MS methods and acquisition LC-MS data; J.W.: development SEM methods and acquiring SEM images; M.E.C.: development Lionheart methods and acquisition Lionheart videos and pictures. G.P.vW. providing *Streptomyces* strains and proofreading; A.R.: conceptualization, supervision, and reviewing. J.H.dW.: conceptualization, supervision, writing, and reviewing.

## Competing interests

The authors declare no competing interests.
