## [Peer Review File · Communications Biology]

Reviewers' comments:

Reviewer #1 (Remarks to the Author):

The paper describes the identification and partial characterization of a conserved esterase (not lipase) active on bis-(2-hydroxyethyl) terephthalate (BHET), which is a frequent intermediate of PET polymer degradation.

The article is well written and timely. While I overall like the idea that moonlighting enzymes (promiscuous) enzymes are involved in PET degradation and that these are induced in their transcription by other naturally polymer degradation products, the current paper is in part preliminary and has few shortcomings.

My main problem with this paper is that the authors have a very interesting BHETase and that they try to make their point on selling it as polymer-active enzyme, which it is not. The polymer activity has only been demonstrated on the SEM images and should be reported on a quantitative way using UHPLC data.

With this respect the current paper lacks in part solid data on overall release of TPA/BHET or MHET from PET foil or PET powder. Since the data imply that the enzyme is not /or only poorly active on the polymer it might be better to shift the focus and sell it as a BHETase which it is. It might be a feruloyl esterase in fact.

Kinetic K_m and k_{cat} data are missing in relation to other 'BHETases'.

The controls in figure 9 and elsewhere should include true BHETases or MHETases from other bacteria. Also LCC has quite some MHETase and BHETase activities. Similarly, the recently published PET46 from Perez-Garcia et al 2023 is a very active BHETase.

The data presented imply that the BHET hydrolysis is 10-fold lower compared to PET46.

Figure 9; I did not understand the rationale behind this figure S2LipA is not PETase; why compare it with PETases and not with other BHETases?

The title appears to be misleading. The enzyme S2LipA is a 'BHETase' and obviously not or only weakly polymer active. Therefore, it would be fair to change the title.

The structure model data are weak, lack in part novelty and imply that it is very similar to the ISPetase. It is however not very similar. It would be better to model it to other closely related esterases with higher similarity. An overall 41% identity with the ISPetase is not very strong and likely results in overinterpretation of active aa. Please be aware these are predicted binding sites and no experimental data on binding have been provided (line 113 onwards). It might be better to align the structure based on a more closely related structure

model.

Introduction: The overall numbers for the production are not correct. It is more in the range of 450 – 500 million tons globally. Lines 28,29 and 37. The better references are here the plastics Europe data.

PET is not the most dominant polymer it is PE.

Line 48 onwards, references are missing.

Yoshida et al did not report on the evolution of a bacterium or enzymes. The simply reported on a bacterium that was able to degrade PET.

There are reports on other enzymes out that have been published prior to the Yoshida paper; see Dresler 2006, Kleeberg 1998; Sulaiman 2012

Line 83.: BHET is a monomer and degradation product of PET. It is not a polymer and therefore no model compound for PET.

Some of the figures are not needed.

The discussion somehow would need to address the point that LipA is a promiscuous esterase (not lipase) enzyme that acts on the feruloyl-like substrates with some specificity. This appears to be a common trait of esterases. See line 407: It is not clear if S2LipA is a lipase or an esterase. I would assume that it is an esterase and I did not see any data implying that the enzyme is a lipase.

Reviewer #2 (Remarks to the Author):

In manuscript “Finding needles in haystacks: identification of novel conserved PETase enzymes in Streptomyces.” by Verschoor et al, they investigated the PET degrading ability of the IsPETase homolog ScLipA from Streptomyces coelicolor, and screened 18% of 96 different Streptomyces strains able to degrade the model substrate BHET. In this study, three different variants, ScLipA, S2LipA, and S92LipA were identified and analyzed the function in the degradation ability of BHET and amorphous PET in Streptomyces species. In addition, they further demonstrated that the knock-out exhibited a significant decrease in its BHET degrading activity by in vivo assay. Finally, the activity on BHET and amorphous PET film was investigated under the optimum pH and temperature, the results showed that S2LipA efficiently degraded BHET and caused roughening and small indents on the surface of PET film. This research presents a full comprehensive identification and characterization

of a novel, conserved PET-degrading enzyme family of Streptomyces. However, the reviewers still have some questions about the manuscript, and a major revision is recommended to address the concerns raised by the reviewers and enhance the manuscript's overall quality for potential publication in Communications Biology.

1. The introduction is considered long and somewhat illogical. The reviewers suggest condensing the first two paragraphs into one. Clearer explanations of the innovation and value of the research are needed. Specify whether ScLipA was the first enzyme of its kind discovered and reported.

Improve the clarity and presentation of figures, ensuring proper formatting of figure legends. In particular, Figures 3A, B, and C are not clear, and marked significant differences may aid reader understanding. Provide the result diagram of Figure 6A again for better clarity.

3. The discussion is criticized for piling up results without in-depth analysis. It's recommended to comprehensively discuss important research results, highlight differences from other studies, and provide brief overviews of future research prospects.

4. Consider using PET waste or PET-derived compounds as substrates in experiments to better reflect real-world conditions of PET waste degradation.

5. Strains' biochemical properties under different culture conditions are questioned. Authors are encouraged to explore if strains have the same biochemical properties under physiological conditions as under Difco agar culture conditions. In addition, provide additional information about GlcNac's impact on strain growth and enzyme production, especially in liquid and solid-state culture conditions.

6. Address the significance of dimerization observed in SDS-PAGE for S92LipA and ScLipA. Explain why they still dimerize and its potential implications for their functions.

Minor:

1. Please unify the writing format of ABCD in the figures for consistency.

2. Consider moving non-core experiments to the Supplement, specifically lines 178-186 regarding agar brand selection.

3. Consider using different colors to distinguish amino acid differences in Figure 1A for

better clarity.

4. Again, Figures 3 and 6 are noted for having low resolution and unclear details. Ensure the clarity and quality of figures for better interpretation.

5. Address errors and clarify descriptions in Figure 8. Ensure consistent writing format in figure descriptions.

6. Address grammatical and clarity issues throughout the manuscript for improved readability.

Reviewer #1 (Remarks to the Author):

The paper describes the identification and partial characterization of a conserved esterase (not lipase) active on bis-(2-hydroxyethyl) terephthalate (BHET), which is a frequent intermediate of PET polymer degradation.

The article is well-written and timely. While I overall like the idea that moonlighting enzymes (promiscuous) enzymes are involved in PET degradation and that these are induced in their transcription by other natural polymer degradation products, the current paper is in part preliminary and has few shortcomings.

1. My main problem with this paper is that the authors have a very interesting BHETase and that they try to make their point on selling it as polymer-active enzyme, which it is not.

We sincerely thank Reviewer #1 for his/her feedback and raising several issues to improve the work described in this manuscript. We indeed were not aiming at selling our new BHETase as a polymer-active enzyme, however, we do understand the point raised and have changed the title of the manuscript to: "Polyester degradation by soil bacteria: identification of novel conserved BHETase enzymes in Streptomyces." In addition, we have diverted the entire focus of the manuscript more clearly towards BHET degradation.

2. The polymer activity has only been demonstrated on the SEM images and should be reported on a quantitative way using UHPLC data. With this respect the current paper lacks in part solid data on overall release of TPA/BHET or MHET from PET foil or PET powder. Since the data imply that the enzyme is not /or only poorly active on the polymer it might be better to shift the focus and sell it as a BHETase which it is. It might be a feruloyl esterase in fact.

Indeed, in the initial version of our manuscript we merely demonstrated degradation activity on PET-film to indicate the potential of the S2LipA variant. We indeed, have not been able to corroborate activity on PET film with consistent HPLC data showing quantitative release of BHET, MHET and TPA. In our comparative approach, as now presented in figure 8, the activity on PET film of the S2LipA variant appears comparable to the TfCut2 cutinase under ambient conditions, confirming a low activity on amorphous PET of this specific variant. We have more clearly explained this in the text to avoid overinterpretation.

3. Kinetic K_m and k_{cat} data are missing in relation to other 'BHETases'. The controls in figure 9 and elsewhere should include true BHETases or MHEtases from other bacteria. Also LCC has quite some MHETase and BHETase activities. Similarly, the recently published PET46 from Perez-Garcia et al 2023 is a very active BHETase. The data presented imply that the BHET hydrolysis is 10-fold lower compared to PET46. Figure 9; I did not understand the rationale behind this figure S2Lipa is not PETase; why compare it with PETases and not with other BHETases?

We appreciate this comment and have now added extensive comparative analysis with several recently described PETase/BHETases: PET40, TfCut2, LCC, IsPETase, and PET46. These enzymes well represent a broad spectrum of BHETases/PETases with each having a different percent identity with the ScLipA variants. We have expressed all enzymes in E. coli and

extensively repeated BHET enzyme assays, resulting in a true comparison of all enzymes as presented in Fig. 1 and Fig. 7.

6. The title appears to be misleading. The enzyme S2LipA is a 'BHETase' and obviously not or only weakly polymer active. Therefore, it would be fair to change the title.

As stated above, we agree with the reviewer that the title should better represent the focus of the current manuscript. We have changed the title of the manuscript to: "Polyester degradation by soil bacteria: identification of novel conserved BHETase enzymes in Streptomyces."

7. The structure model data are weak, lack in part novelty and imply that it is very similar to the IsPetase. It is however not very similar. It would be better to model it to other closely related esterases with higher similarity. An overall 41% identity with the IsPetase is not very strong and likely results in overinterpretation of active aa. Please be aware the these are predicted binding sites and no experimental data on binding have been provided (line 113 onwards). It might be better to align the structure based on a more closely related structure model.

As mentioned above, we have extended our comparison and included alignments of the ScLipA variants with PET40, TfCut2, LCC, and IsPETase. We have now included structural comparison with PET40 which is a clearly more related BHETase enzyme with 78% identity as presented in Fig.1 . To avoid overstating of the binding site conservation, we have changed line 98 to: "The catalytic triad was well-conserved, as well as two of the three residues of the previously predicted substrate binding site [38]"

8. Introduction: The overall numbers for the production are not correct. It is more in the range of 450–500 million tons globally. Lines 28, 29 and 37. The better references are here the plastics Europe data. PET is not the most dominant polymer it is PE.

We agree with the reviewer that PE on a global scale is a more dominant polymer than PET and have changed the production numbers mentioned in lines 40, 41 and 45 to the numbers corresponding with the latest plastics Europe numbers, with adequate reference.

9. Line 83.: BHET is a monomer and degradation product of PET. It is not a polymer and therefore no model compound for PET.

We of course agree with the reviewer about this notion. Line 57 was rewritten as: "Where the genus Thermobifida contains approximately twenty enzymes with depolymerizing activity for PET- or its constituent oligomer bis(2-hydroxyethyl) terephthalate (BHET), for various other genera including Streptomyces only limited information is available [11], [12], [13], [14], [15], [16])."

10. Some of the figures are not needed.

After careful consideration we agree with reviewer #1 and reviewer #2 who both made this same remark. We have now moved several Figures to the supplementary data; original Figures 2 and 3 are partially combined and partially moved to the supplemental data as supplement 1,2 and 3. Additionally, figure 5 is now are included as Supplementary data S7 respectively.

11. The discussion somehow would need to address the point that LipA is a promiscuous esterase (not lipase) enzyme that acts on the feruloyl-like substrates with some specificity. This appears to be a common trait of esterases. See line 407: It is not clear if S2LipA is a lipase or an esterase. I would assume that it is an esterase and I did not see any data implying that the enzyme is a lipase.

We agree with reviewer 1 and after consideration we have concluded that S2LipA as an promiscuous esterase; Therefore we propose renaming to a BHET degrading esterase (Bde) in Line 385-386. "Therefore, we suggest re-evaluating the annotation of those genes and enzymes and changing it to BHET degrading esterase A, or BdeA."

Reviewer #2 (Remarks to the Author):

In manuscript “Finding needles in haystacks: identification of novel conserved PETase enzymes in *Streptomyces*.” by Verschoor et al, they investigated the PET degrading ability of the IsPETase homolog ScLipA from *Streptomyces coelicolor*, and screened 18% of 96 different *Streptomyces* strains able to degrade the model substrate BHET. In this study, three different variants, ScLipA, S2LipA, and S92LipA were identified and analyzed the function in the degradation ability of BHET and amorphous PET in *Streptomyces* species. In addition, they further demonstrated that the knock-out exhibited a significant decrease in its BHET degrading activity by in vivo assay. Finally, the activity on BHET and amorphous PET film was investigated under the optimum pH and temperature, the results showed that S2LipA efficiently degraded BHET and caused roughening and small indents on the surface of PET film. This research presents a full comprehensive identification and characterization of a novel, conserved PET-degrading enzyme family of *Streptomyces*. However, the reviewers still have some questions about the manuscript, and a major revision is recommended to address the concerns raised by the reviewers and enhance the manuscript's overall quality for potential publication in *Communications Biology*. –

We sincerely thank Reviewer #2 for his/her constructive feedback and raising several issues to improve the work described in this manuscript.

Major

1. The introduction is considered long and somewhat illogical. The reviewers suggest condensing the first two paragraphs into one. Clearer explanations of the innovation and value of the research are needed. Specify whether ScLipA was the first enzyme of its kind discovered and reported.

*We have now shortened the introduction significantly and have tried to focus more clearly on the novelty of our findings. Accordingly, we have added the novelty of our findings in Line 69: “For the entire genus of *Streptomyces*, only three PHEs have been described, namely SM14est, Sub1 and PET40 [12], [30], [31]. Hence, limited data is available concerning the ability of *Streptomyces* to degrade BHET and PET and thus on the emergence, distribution, conservation, and activity of their corresponding BHET-degrading enzymes.”*

2. Improve the clarity and presentation of figures, ensuring proper formatting of figure legends. In particular, Figures 3A, B, and C are not clear, and marked significant differences may aid reader understanding. Provide the result diagram of Figure 6A again for better clarity.

We understand the point the reviewer raises. We have redesigned Figure 3 for more clarity. However, indicating significance might increase confusion in some figures. For example, in figure 2C, 2D, 4C and F we try to show the presence of three compounds it is difficult to show the significant difference of all compounds without creating a confusing figure. Therefore, we decided not to indicate the significance in the figure, the statistical analysis is discussed in the text, materials and methods and is provided as supplementary data S9. For figure 2D and 8D we decided to focus on the significance of BHET reduction. For figure 7, the enzymatic degradation the significant difference with the buffer was displayed as indicated in the figure legend.

3. The discussion is criticized for piling up results without in-depth analysis. It's recommended to comprehensively discuss important research results, highlight differences from other studies, and provide brief overviews of future research prospects.

We have rewritten the discussion significantly with the comments and suggestions from both reviewers in mind. We believe the discussion has significantly improved and describes the novel BHETases of Streptomyces in the right comprehensive context.

4. Consider using PET waste or PET-derived compounds as substrates in experiments to better reflect real-world conditions of PET waste degradation.

The ScLipA esterases are promiscuous enzymes with clear BHET-degrading activity, with the S2LipA variant exhibiting minor activity on PET-film. Hence, we do not want to claim efficient PET-degradation by the Streptomyces enzymes and have now clarified our focus on BHETase activity and comparison with other BHETases. Specifically, since activity of the ScLipA enzymes on amorphous PET is low we do certainly not expect degradation of post-consumer PET waste. For such purpose, extensive enzyme engineering and improvement will be required, as stated in the discussion.

5. Strains' biochemical properties under different culture conditions are questioned. Authors are encouraged to explore if strains have the same biochemical properties under physiological conditions as under Difco agar culture conditions. In addition, provide additional information about GlcNac's impact on strain growth and enzyme production, especially in liquid and solid-state culture conditions.

Since we wanted to maintain focus on the comparison of enzyme activities but did not want to conceal the effect of different growth media on strain growth and enzyme activity, we have decided to move details concerning the agar brand selection to the supplementary data. Moreover, we have added additional explanation concerning the effects of GlcNac on strain development and enzyme production in accordance with available literature, in Lines 121-126, 170-173 and 352-354.

6. Address the significance of dimerization observed in SDS-PAGE for S92LipA and ScLipA. Explain why they still dimerize and its potential implications for their functions.

We understand the question of the reviewer, however, the occurrence of the putative dimer on Western blot does not seem to be consistent. Moreover, activity seems to reside with the monomeric bands. Therefore, we cannot make any conclusions regarding the dimer, as described in lines 279-284.

Minor:

1. Please unify the writing format of ABCD in the figures for consistency.

We appreciate this comment from reviewer #2 and have adapted Figures accordingly.

2. Consider moving non-core experiments to the Supplement, specifically lines 178-186 regarding agar brand selection.

As stated above we have moved several experiments to the supplement to improve focus and clarity of the manuscript, including details concerning the agar brand selection.

3. Consider using different colors to distinguish amino acid differences in Figure 1A for better clarity.

We have now added colors to better visualize the differences.

4. Again, Figures 3 and 6 are noted for having low resolution and unclear details. Ensure the clarity and quality of figures for better interpretation.

We appreciate the comment of reviewer #2, we believe the low resolution is due to compressing the document. All images were exported from Illustrator or Indesign in the highest possible quality, this resulted in very large images which made the document too large to submit. To avoid this we will also provide the images in a separate zip file.

5. Address errors and clarify descriptions in Figure 8. Ensure consistent writing format in figure descriptions.

We have checked all figure descriptions and are confident that the writing format is now consistent.

6. Address grammatical and clarity issues throughout the manuscript for improved readability.

We have again reviewed the entire text for any typos and grammatical errors and are confident that the text is consistent overall.

REVIEWERS' COMMENTS:

Reviewer #1 (Remarks to the Author):

This is a revised version of a previously submitted manuscript and describing a novel BHETase.

The authors have addressed most of my concerns and rewritten the manuscript largely. They shifted the focus and also included significant number of current literature on this topic. The manuscript has greatly improved and is now in an excellent shape.

I have no further objections and concerns.

Reviewer #2 (Remarks to the Author):

I have thoroughly reviewed the revised version of your manuscript, and I must commend the significant effort the authors have put into addressing the concerns and suggestions raised in my previous review. The revisions have notably enhanced the clarity and comprehension of the content, reflecting a commendable dedication to improving the manuscript. However, despite these improvements, there remain a few critical areas that necessitate further attention before the manuscript can be considered ready for publication.

1. The current title of the article incorporates the term "BHETase" which, upon reflection and in alignment with the content of your study, might be more accurately described using the original term "PETase". Moreover, the descriptor "novel conserved PETase enzymes" seems to overstate the innovation of the study. Given the context and findings, a more accurate representation would be "conserved PETase-like enzymes", thereby necessitating the removal of the term "novel" from the title to ensure precision and honesty in reporting.

2. A pivotal concern is the clear definition and distinction between BHETase and PETase within your study. I strongly recommend a thorough examination of the literature, particularly the study titled "Discovery and mechanism-guided engineering of BHET hydrolases for improved PET recycling and upcycling". This reference will provide valuable insights into the differentiation between the enzymes, especially considering that, while PETase also exhibits BHETase catalytic activity, the enzymes investigated in your study are more appropriately categorized as PETase-like based on their structural and catalytic mechanisms. This distinction is crucial for the accuracy and integrity of your work.

3. The modifications made to the figures, particularly Figure 1, warrant a reevaluation. It is suggested that the original Figure 1 might offer clearer and more effective visualization of the concepts being presented. Specifically, the structure depicted in the revised Figure 1 could benefit from reverting to the original versions of Figures 1D, 1E, 1F (with the latter three being consolidated into a new Figure 1B), which should replace the current Figures 1B, 1C.